# Concomitant Endoscopic Surgery for Bladder Tumors and Prostatic Obstruction: Are We Safely Hitting Two Birds with One Stone? A Systematic Review and Meta-Analysis

**DOI:** 10.3390/jcm11206208

**Published:** 2022-10-21

**Authors:** Eliophotos Savvides, Nikolaos Pyrgidis, Georgios Langas, Evangelos N. Symeonidis, Georgios Dimitriadis, Petros Sountoulides

**Affiliations:** 11st Department of Urology, Faculty of Health Sciences, Medical School, Aristotle University of Thessaloniki, GR-55134 Thessaloniki, Greece; 2Department of Urology, ‘Martha-Maria’ Hospital Nuremberg, 90402 Nuremberg, Germany

**Keywords:** bladder cancer, benign prostatic obstruction, concomitant, transurethral resection, recurrence

## Abstract

Background: Lower urinary tract symptoms (LUTS) caused by benign prostatic obstruction (BPO) and bladder tumors may co-exist, especially among elderly patients. Transurethral resection of bladder tumors (TURBT) and endoscopic surgery for benign prostatic obstruction in the same setting are avoided by many surgeons due to concerns for tumor cell seeding and recurrences in the prostatic urethra. Aim: The aim of this study was assess the effect of concomitant TURBT and endoscopic BPO surgery on oncological safety and patient quality of life via systematic review and meta-analysis. Methods: We searched the PubMed, Cochrane Library, EMBASE, Scopus, and Clinicaltrials.gov databases and sources of grey literature published before June 2021 for relevant studies. We performed a random-effects meta-analysis of odds ratios (ORs) or weighted mean differences (WMD) to compare concomitant TURBT and BPO surgery versus TURBT alone in terms of recurrence and progression rates. Accordingly, we undertook multiple subgroups and sensitivity analyses (PROSPERO: CRD42020173363). Results: Three randomized and twelve retrospective observational studies with 2421 participants were included. Across studies with good methodological quality, no statistically significant differences were demonstrated regarding overall bladder tumors recurrence rates between concomitant endoscopic BPO surgery and TURBT versus TURBT alone (OR: 0.81, 95% CI: 0.60–1.09, I^2^ = 42%). Similarly, no significant differences were observed in recurrences located at the bladder neck and/or prostatic urethra (OR: 1.06, 95% CI: 0.76–1.47, I^2^ = 0%), time to first recurrence (WMD: −0.2 months, 95% Cl: −2.2–1.8, I^2^ = 48%), and progression rate (OR: 1.05, 95% CI: 0.67–1.64, I^2^ = 0%). Subgroup analyses based on tumor grade, number of tumors, and utilization of single-instillation chemotherapy post-TURBT did not detect any significant differences in overall bladder tumor recurrence. The level of evidence was estimated as low for all outcomes. Concomitant surgery improved lower urinary tract symptoms. Conclusion: Concomitant endoscopic BPO surgery and TURBT are oncologically safe and improve LUTS-related quality of life.

## 1. Introduction

Bladder cancer, with worldwide age-standardized incidence rates of (per 100,000 person/y) 9.0 for men and 2.2 for women, diagnosed in 75% of cases as non-muscle-invasive bladder cancer (NMIBC), is managed by transurethral resection of bladder tumors (TURBT) and adjuvant intravesical immunotherapy or chemotherapy, depending on patient and tumor characteristics [1,2,3]. Patients diagnosed with NMIBC have a 5-year recurrence risk of 31–78% and a 5-year progression risk of 0.8–45% [4], depending on tumors stage and grade, adding a significant burden for both patients and healthcare system resources [5]. Accordingly, benign prostatic obstruction (BPO), highly prevalent among elderly men [6], impairs quality of life and is surgically managed [7], in most cases, by endoscopic transurethral prostate surgery [8]. Specifically, BPO prevalence among men over the age of 50 years is 50–75%; over the age of 70 years, 80% on average are impacted; and over the age of 80 years, prevalence is almost 90% [1]. It is estimated that approximately 6% of patients diagnosed with NMIBC will have simultaneous BPO [9]. In some cases, bladder tumors are incidentally detected during elective transurethral surgery for BPO. In most cases, bladder tumors are diagnosed in patients with either absolute (retention, recurrent hematuria, etc.) or relative indications for BPO surgery (worsening of LUTS despite medical therapy). In this scenario, clinicians avoid performing simultaneous TURBT and transurethral prostatic surgery mostly for fear of cancer cells seeding in the prostatic urethra or bladder neck [10,11,12]. The EAU guidelines suggest, albeit based on low-grade evidence, that resection of small papillary bladder tumors may be safely performed during transurethral prostate surgery. This recommendation derives predominantly from evidence from two recent systematic reviews and meta-analyses demonstrating that concomitant TURBT and TURP do not increase the recurrence and progression rate of NMIBC [13,14].

However, the clinical studies powering these meta-analyses were tempered by a lack of evidence regarding the effect of different types of transurethral prostate surgery (laser, electrocautery, enucleation) and the impact of BPO treatment on lower urinary tract symptoms (LUTS) and quality of life. Additionally, the authors did not adjust for risk factors such as multifocality and grade of NMIBC, which might have affected outcomes. Within this scope, and aiming to provide solid evidence, we generated a systematic review and meta-analysis to assess the effect of concomitant TURBT and endoscopic prostatic surgery on oncological outcomes and patients’ quality of life.

## 2. Materials and Methods

### 2.1. Search Strategy

The objectives and methods of our study were predefined and documented in the study protocol registered at PROSPERO (ID: CRD42020173363). This systematic review and meta-analysis were performed according to the Preferred Reporting Items for Systematic reviews and Meta-Analyses (PRISMA) statement [15]. Two independent reviewers (E.S., G.L.) systematically searched Medline via PubMed, Scopus, Cochrane Library, EMBASE, and Clinicaltrials.gov databases for relevant studies until June 2021. The search terms comprised [(prostate) AND (bladder cancer) OR (bladder tumor) OR (bladder neoplasm) OR (bladder carcinoma)], as well as relevant synonyms, truncated words, and MeSH terms (Appendix A). Sources of grey literature, including conference abstracts published in major urology journals, were hand-searched, and the reference lists of all eligible studies and relevant reviews were perused.

### 2.2. Study Eligibility Criteria

Our inclusion criteria comprised randomized controlled trials (RCTs), prospective or retrospective comparative studies published in any language, enrolling patients with NMIBC and BPO who underwent transurethral prostate surgery, and TURBT versus TURBT alone. On the contrary, we excluded case reports and case series, as well as single-arm, animal, or cadaveric studies. When records with overlapping patient populations were identified, we only considered the most recent study.

### 2.3. Data Extraction and Quality Assessment

Two independent reviewers (E.S., G.L.) separately assessed the titles and abstracts of all identified titles. Full texts of all eligible studies were screened according to the predefined selection criteria. A predefined Microsoft Excel spreadsheet was developed in consultation with all authors to extract relevant information about the study and patient characteristics (author, design, follow-up period, and patient age), bladder tumor characteristics (stage, grade, number of tumors, size, and concomitant carcinoma in situ), intervention details (adjuvant therapy, indication for concomitant surgery), and outcomes (overall and bladder neck/prostatic urethra recurrence rate, progression rate, LUTS, and quality of life). Regarding histological grading of NMIBC, given that some studies used the WHO 1973 (G1–G3) grading system while others used the WHO 2004/2016 system, we considered grade 1 tumors as low-grade, while grade 2 and 3 tumors were considered as high-grade. Any disagreements were resolved by a third reviewer (N.P.).

Quality assessment of RCTs was performed in duplicate by two reviewers (E.S., G.L.) using the Cochrane Collaboration’s Risk of Bias Tool version 2 (RoB-2) [16]. For non-RCTs, the same reviewers assessed all studies with a modified version of the Newcastle Ottawa Scale (NOS), and any discrepancies were resolved by a third reviewer (N.P.). Each non-RCT was judged by items grouped into three distinct categories (selection of study groups, comparability of groups, and outcomes). The modified NOS version is presented in Appendix A.

### 2.4. Grading of Evidence, Data Synthesis, and Statistical Analysis

We provided a qualitative synthesis regarding the impact of concomitant transurethral prostate surgery versus TURBT alone on LUTS and quality of life. Additionally, we performed a meta-analysis of odds ratios (ORs) for categorical outcomes and a meta-analysis of weighted mean differences (WMD) for continuous outcomes using an inverse variance random-effects model for studies comparing concomitant TURBT and transurethral prostate surgery versus TURBT alone. In particular, we synthesized meta-analytic effects with the corresponding confidence intervals (CIs) for: (i) risk of bladder tumors recurrence and mean time to recurrence; (ii) bladder neck and prostatic urethra tumors recurrence; (iii) unifocal or multifocal bladder tumors recurrence; (iv) recurrence at low- or high-grade NMIBC; (v) progression to muscle-invasive bladder cancer.

Furthermore, we performed subgroup analyses according to study type (RCTs versus non-RCTs), as well as based on the use of adjuvant postoperative intravesical chemotherapy. Accordingly, we undertook a sensitivity analysis, including studies at low risk of bias. Heterogeneity for all outcomes was evaluated with the I^2^, and its significance was estimated based on the *p*-value of Cochran’s Q test [17]. Additionally, we explored potential publication bias via visual assessment of funnel plot asymmetry and Egger’s statistical test [18]. Of note is that we applied the Grading of Recommendations Assessment, Development, and Evaluation (GRADE) approach in addressing the overall strength of evidence for all outcomes [19]. More specifically, two reviewers (E.S., G.L.) graded the risk of bias, inconsistency, indirectness, imprecision, and publication bias among the included studies for all meta-analytic effects. For all estimations, *p*-values lower than 0.05 were considered statistically significant. All analyses were performed with the R statistical software, version 3.6.3, (R Foundation for Statistical Computing, Vienna, Austria) using the “meta” package.

## 3. Results

### 3.1. Study Results and Quality Assessment

The literature search yielded 1667 potentially relevant articles. After title and abstract screening, 29 articles underwent full-text evaluation. Ultimately, 15 studies with 2421 participants were included in the qualitative and quantitative synthesis (3 RCTs and 12 retrospective observational studies) [20,21,22,23,24,25,26,27,28,29,30,31,32,33,34]. The PRISMA flow chart of the study-selection process is illustrated in Figure 1, and all excluded studies, along with reasons for exclusion, are presented in Appendix A.

Across included studies, the mean patient age ranged from 57 to 71 years of age, while the mean follow-up ranged from 29 to 52 months. Two RCTs compared concomitant TURBT and TURP versus TURBT and medical therapy for BPO [21,31], whereas the remaining studies compared TURBT and transurethral prostate surgery versus TURBT alone. In particular, in 13 studies, BPO was managed by TURP, while in 1 study, it was managed with photo-selective vaporization of the prostate and in another it was managed with thulium laser enucleation [28,34]. Interestingly, in those two studies where BPO was not managed with TURP, postoperative single-instillation intravesical chemotherapy was administered.

Overall, a total of 1041 patients underwent concomitant surgery. Of them, 489 patients (47%) presented with bladder tumors and known BPO; in 45 (4.5%), concomitant TURP was undertaken to enable access to bladder tumors at certain locations within the bladder; and 16 (1.5%) underwent TURBT during elective surgery for BPO due to incidental finding of bladder tumors. In the remaining 491 cases (47%), the reason for concomitant surgery was not reported. The baseline characteristics of all included studies are displayed in Table 1.

Based on the RoB-2 tool, all RCTs displayed some methodological concerns. Based on the NOS, eight non-RCTs were considered of good quality [24,26,27,29,30,32,33,34], three of moderate quality [20,22,23], and one of poor quality [25]. The detailed quality assessment is depicted in Appendix A.

### 3.2. Bladder Tumor Recurrence

All studies provided comparative data about bladder tumor recurrence rate in concomitant surgery versus TURBT alone. Within a mean follow-up of 50.7 months (range: 26.8–96 months), 400 of 1041 patients (38.4%) that underwent concomitant TURBT and prostatic surgery developed bladder tumors recurrence compared to 554 of 1380 patients (40.1%) in the TURBT-only arm. The analysis of data displayed a statistically significant difference in favor of concomitant treatment (OR: 0.78, 95% CI: 0.64–0.95, I^2^ = 14%, Prediction interval: 0.54 to 1.14, Figure 2). Still, in the sensitivity analysis of studies with good methodological quality, no statistically significant differences were demonstrated between concomitant surgery and TURBT alone (OR: 0.81, 95% CI: 0.6–1.09, I^2^ = 42%, Appendix A).

When comparing studies where postoperative single intravesical instillation of chemotherapy was given versus no intravesical chemotherapy, no significant differences were observed (*p* = 0.99, Appendix A). Similarly, the mean time to the first recurrence was not significantly different between the two groups (WMD: −0.2 months, 95% Cl: −2.2–1.8, I^2^ = 48%, Appendix A). Of interest is that the funnel plot inspection and Egger’s statistical test did not indicate important publication bias among the included studies (Appendix A).

All included studies provided data on bladder tumor recurrence at the bladder neck or the prostatic urethra. At a mean follow-up of 50.7 months (range: 26.8–96 months), a bladder neck or prostatic urethra recurrence occurred in 84 of 1041 cases (8.1%) after concomitant surgery versus 87 of 1380 cases (6.3%) after TURBT alone without any significant difference observed between the two groups (OR: 1.06, 95% CI: 0.76–1.47, I^2^ = 0%, Figure 3). Similarly, in the subgroup analysis for single intravesical instillation of chemotherapy, no significant difference was detected among studies (*p* = 0.75, Appendix A).

### 3.3. Bladder Tumor Recurrences Depending on Tumor Grade and Number of Tumors

Three studies (with 687 patients) provided information on recurrences based on tumor grade [23,29,33]. No difference was seen in tumor recurrence rates following concomitant surgery versus TURBT alone (Appendix A) for low-grade (OR: 0.94, 95% CI: 0.47–1.87, I^2^ = 30%) and high-grade tumors (OR: 0.86, 95% CI: 0.54–1.35, I^2^ = 0%), Additionally, three studies (700 participants) provided information on recurrences based on the number of tumors at initial TURBT (Appendix A) [23,24,29]. For both unifocal (OR: 0.69, 95% CI: 0.37–1.29, I^2^ = 24%) and multifocal tumors (OR: 0.77, 95% CI: 0.43–1.41, I^2^ = 58%), no differences were demonstrated in tumor recurrence rates after concomitant surgery versus TURBT alone.

### 3.4. Bladder Tumor Progression

Nine studies recruiting 1254 participants reported relevant data on the progression of NMIBC to muscle-invasive bladder cancer [20,24,26,27,28,30,31,32,34]. No significant differences were observed between the two groups, with 40 of 528 patients (7.6%) in the concomitant and 51 of 526 patients (9.7%) in the TURBT group developing tumor progression to muscle-invasive during follow-up (OR: 1.05, 95% CI: 0.67–1.64, I^2^ = 0%, Prediction interval: 0.61–1.79, Figure 4). In the subgroup analysis for single intravesical instillation of chemotherapy, no significant differences were detected among the studies (*p* = 0.46, Appendix A).

### 3.5. Lower Urinary Tract Symptoms

A total of four studies (346 participants) assessed the effect of concomitant TURB and TURP versus TURBT alone on LUTS and quality of life both pre- and postoperatively [21,24,31,34]. Singh et al. argued that concomitant TURBT and TURP resulted in a marked, although indirect, improvement of patient’s quality of life by improving the International Prostate Symptoms Score (IPSS) by approximately 70–80% without using a dedicated quality-of-life-evaluation tool. Accordingly, Ham et al. and Wang et al. compared uroflowmetry data for concomitant TURBT and TURP versus TURBT alone. As expected, significant improvement in post-void residual and maximum flow rate was reported after concomitant surgery compared to TURBT alone. The authors concluded that concomitant TURBT and TURP might ameliorate patients’ quality of life by improving LUTS. Of interest, in the only study exploring the effect of concomitant TURBT and TURP on quality of life as the primary outcome, the authors demonstrated that both the International Prostate Symptoms Score and the Functional Assessment of Cancer Therapy score had significantly improved.

### 3.6. Quality of Evidence

We assessed the quality of evidence for all outcomes separately based on study design (RCT and non-RCT). Even though the significance of all outcomes was deemed critical, the quality of evidence was considered low. The retrospective design and the lack of adequate adjustment for confounding factors among non-RCTs, as well as the lack of allocation concealment, absence of blinding, and the relatively small sample size among RCTs, downgraded the quality of evidence for all outcomes (Appendix A).

## 4. Discussion

Concomitant TURBT and prostatic surgery have not been routine clinical practice as it has been postulated that disruption of the prostatic urethra urothelial lining caused by surgery may lead to bladder cancer cells seeding in the traumatized prostatic urethra. This theory was introduced by Albarran and Imbert while discussing the risks of urothelial cancer recurrence after renal pelvic tumor excision [10,12,35]. In support of the tumor-seeding theory, a recent meta-analysis suggested that prophylactic stenting after the resection of tumors involving the bladder orifice should be avoided as it increases the risk of metachronous upper tract urothelial cancer [36].

On the other hand, there are studies suggesting that resolution of BPO reduces tumor recurrence by limiting the interaction between urine carcinogens with susceptible bladder epithelium [37], proposing that concomitant surgery does not negatively affect oncological outcomes and may be a safe and feasible option in selected cases [3,13,14,38,39].

Until recently, the dilemma of whether endoscopic surgery for bladder tumors and BPO can be performed in the same sitting without compromising oncological safety by increasing the risk of tumor cell re-implantation and recurrence had not been resolved. The 2021 version of the EAU guidelines recommended concomitant prostate and bladder surgery if bladder tumors are papillary, small, and singular. This recommendation was mainly based on the results of a recent meta-analysis by Motlagh et al. [14], demonstrating the absence of an increased risk of the bladder neck or prostatic urethra recurrence following concomitant TURBT and endoscopic prostate surgery compared to TURBT.

The present systematic review and meta-analysis reinforce the evidence suggesting that concomitant prostatic surgery does not lead to increased bladder tumor recurrence rates compared to TURBT alone in patients with NMIBC. Our findings are more or less in line with the meta-analysis by Motlagh et al. [14]. However, we have provided included recent publications and grey literature studies, thus accumulating a bigger sample and reducing publication bias. Furthermore, we have implemented GRADE in our findings, defining their overall strength.

Moreover, our meta-analysis has provided additional evidence to that presented by Motlagh et al., as we have performed certain critical subgroup analyses looking at the role of tumor grade, multifocality, and adjuvant intravesical chemotherapy on oncological safety in the setting of concomitant surgery. The time to first recurrence and the recurrence rate of tumors in the bladder, neck, or prostatic urethra do not differ between patients undergoing concomitant surgery versus patients undergoing TURBT only. Similarly, the presence of multifocal and high-grade tumors does not seem to increase the recurrence rate indicating that patients with multifocal and high-grade tumors should not be excluded from concomitant prostate surgery.

Single intravesical instillation of chemotherapy is recommended for most patients undergoing TURBT as it reduces tumor recurrences by destroying circulating tumor cells and displaying an ablative effect on residual tumor cells [40]. However, in line with the low adherence to single-instillation chemotherapy [41], postoperative instillation across included studies was performed only in studies where prostate surgery was performed by laser and not by electrosurgery [28,34]. Single-instillation postoperative chemotherapy following TURP was avoided due to the increased risk of bleeding and extravasation of the chemotherapeutic agent. Even though no significant difference was detected in the recurrence rate after single instillation intravesical chemotherapy versus no chemotherapy, robust evidence on this issue is lacking. Therefore, further studies exploring the application of mitomycin or other intravesical chemotherapy in this setting are needed.

Another possible confounding factor in our analysis was the lack of reporting of BCG instillations. Intravesical instillation of immunotherapy, in the form of BCG, activates the immune system with the possible involvement of bladder, and cancer cells, including the attachment and the internalization of BCG, locoregional secretion of cytokines and chemokines, and presentation of BCG and/or cancer cell antigens to cells of the immune system. Immune system cell subsets that have potential roles in BCG therapy include CD4+ and CD8+ lymphocytes, natural killer cells, granulocytes, macrophages, and dendritic cells [42]. Different BCG instillation regimens are proposed according to the risk stratification, aiming to prevent the tumor’s recurrence and progression.

The present analysis is the first to provide evidence of patients’ quality of life following concomitant TURBT and prostate surgery. Concomitant surgery improves LUTS and, therefore, ameliorates patient quality of life. Still, it should be highlighted that the robustness of our findings was considered low for all outcomes, as the available evidence derived predominantly from retrospective observational studies that enrolled patients with different tumor characteristics and, thus, did not adequately adjust for potential confounding factors.

After drafting our paper, a new multicentric observational study comparing TURB vs. TURBT and TURP, as far as recurrences and disease progression, was published [43]. The study population, spread across 12 European hospitals, consisted of 581 patients that underwent TURB with 181 in the simultaneous arm. Confounds factors were examined thoroughly, and subgroup analysis according to risk stratification was performed. Interestingly enough, the simultaneous group had a statistically significant difference in both recurrence and recurrence-free survival (RFS). More precisely, TURBT and TURP groups displayed a 28% recurrence rate in comparison with 47% in the TURB group (*p* < 0.001), and 3-year estimates for RFS were in favor of the simultaneous group. These findings are in line with our results, proposing that concomitant surgery is oncologically safe.

There is variation in the clinical management of simultaneous NMIBC and BPO, given the lack of strong recommendations in favor of concomitant surgery. In this context, we hope to have provided a systematic review and meta-analysis exploring, in a holistic approach, the effect of concomitant TURBT and transurethral prostate surgery versus TURBT alone by addressing multiple outcomes and performing subgroup and sensitivity analyses. We have also estimated prediction intervals for all outcomes suggesting that, in future studies, the implementation of concomitant surgery does not aggravate recurrence or progression rates. Furthermore, given that NMIBC and BPO add a heavy burden on healthcare costs [44,45], we have explored the effect of concomitant surgery on patient quality of life. Of note is that, despite our strict eligibility criteria, we have included the largest number of studies and participants compared to relevant meta-analyses and estimated the oncological outcomes after concomitant surgery compared to TURBT alone, adjusting for the number of tumors, grade, and chemotherapy application.

Despite our efforts, the findings of the present study were mitigated by limitations relevant to the retrospective design of most studies, the relatively short and different follow-up regimens, and the rather small number of included studies. Because we did not have access to patient-level data, we could not adjust for specific risk factors and comorbidities, which might have affected outcomes. In particular, data on cancer characteristics, for example, tumor grade, tumor multifocality, history of previous bladder tumors, concomitant carcinoma in situ, and adjuvant BCG instillations, remained under-reported in most of the included studies. Therefore, it should be stressed that future RCTs or well-designed observational studies should adjust for these potential confounders.

## 5. Conclusions

Patients with concomitant NMIBC and BPO that undergo TURBT and prostatic surgery display similar overall and bladder neck or prostatic urethra recurrence and progression rates compared to TURBT alone. Multifocal and high-grade tumors do not seem to increase the recurrence rate after concomitant surgery versus TURBT alone. Concomitant surgery improves LUTS and, therefore, indirectly ameliorates patients’ quality of life. However, it should be highlighted that the level of provided evidence on this issue is low. Therefore, further high-quality, high-volume prospective studies are mandatory to corroborate our findings.

## Figures and Tables

**Figure 1 jcm-11-06208-f001:**
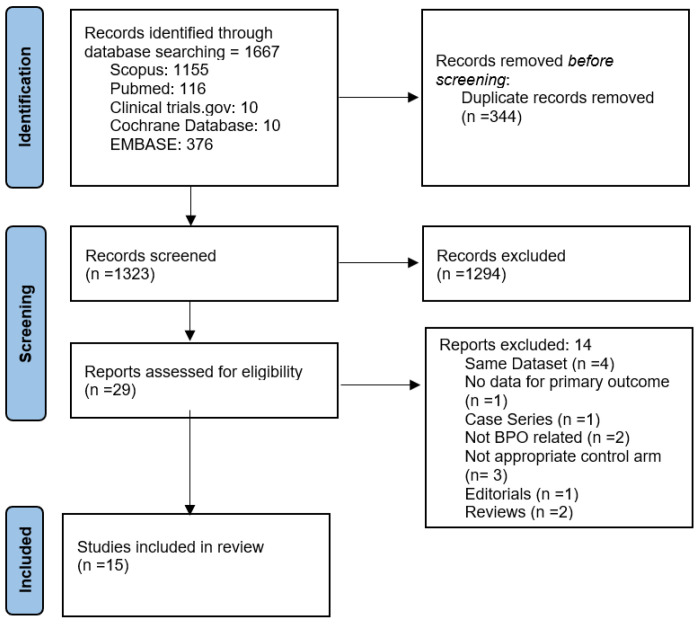
Prisma flow diagram of study-selection process.

**Figure 2 jcm-11-06208-f002:**
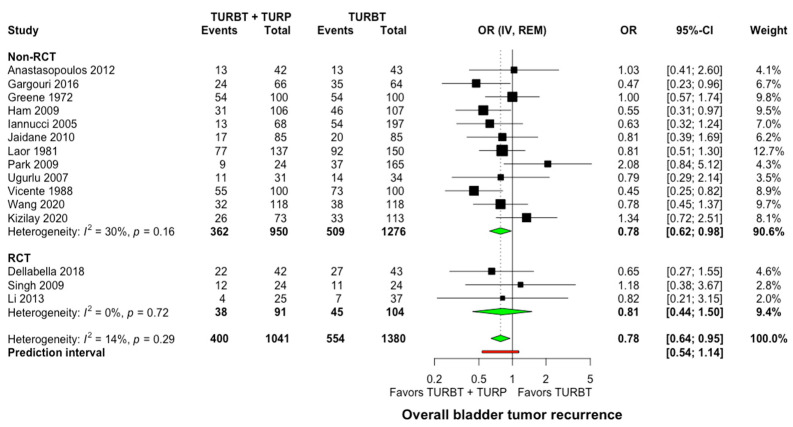
Forest plot comparing concomitant TURBT and TURP versus TURBT in terms of bladder tumor recurrence. CI: confidence interval; IV: inverse variance; OR: odds ratio; RCT: randomized controlled trial; REM: random-effects model; TURBT: transurethral resection of bladder tumor; TURP: transurethral resection of prostate.

**Figure 3 jcm-11-06208-f003:**
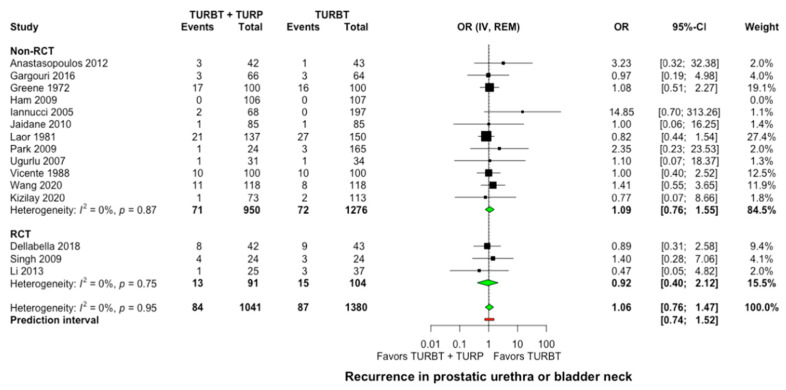
Forest plot comparing concomitant TURBT and TURP versus TURBT in terms of prostatic urethra and bladder neck recurrence. CI: confidence interval; IV: inverse variance; OR: odds ratiο; RCT: randomized controlled trial; REM: random-effects model; TURBT: transurethral resection of bladder tumor; TURP: transurethral resection of prostate.

**Figure 4 jcm-11-06208-f004:**
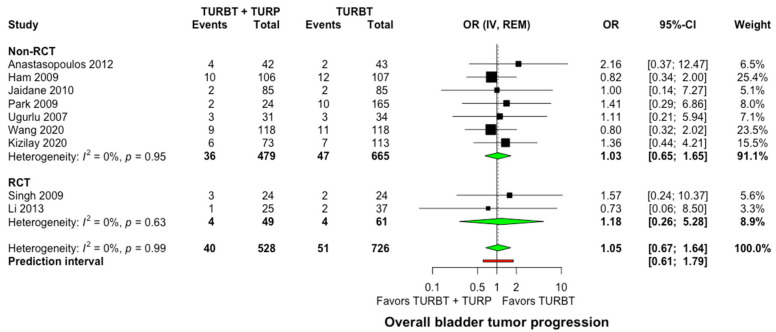
Forest plot comparing concomitant TURBT and TURP versus TURBT in terms of bladder tumor progression. CI: confidence interval; IV: inverse variance; OR: odds ratiο; RCT: randomized controlled trial; REM: random-effects model; TURBT: transurethral resection of bladder tumor; TURP: transurethral resection of the prostate.

**Table 1 jcm-11-06208-t001:** Characteristics of all included studies.

Author, Year of Publication, Study Design	Sample Size (SG/CG)	Mean Age (Years) (SG/CG)	Prostatic Procedure	BC Stage (SG/CG)	Tumor Grade (SG/CG)	Unifocal Tumor (SG/CG)	Multifocal Tumor (SG/CG)	Mean Follow-Up (Months) (SG/CG)	Adjuvant Therapy (SG/CG)	LUTS Assessment
**Anastasopoulos, 2012 [20]** **Non-RCT**	85 (42/43)	69/69	Transurethral resection	NA	NA	ΝA	ΝA	31/27	ΝA	NA
**Dellabella, 2018 [21]** **RCT**	85 (42/43)	64/63	Transurethral resection	pTa/pTis: 29/26 pT1: 13/17	Low: 16/14 High: 26/29	21/24	21/19	37/35	BCG: 27/26	IPSS, Qmax, PVR, FACT-BIQ
**Gargouri, 2016 [22] Non-RCT**	130 (66/64)	72/64	Transurethral resection	NA	NA	38/26	28/38	41/42	BCG: 26/40	NA
**Greene, 1972 [23] Non-RCT**	200 (100/100)	NA	Transurethral resection	NA	G1: 57/59 G2: 29/23 G3: 14/18	81/77	19/23	NA	NA	ΝA
**Ham, 2009 [24]** **Non-RCT**	213 (106/107)	68/66	Transurethral resection	pTa: 21/19 pT1: 85/88	Low: 60/59 High: 46/48	58/56	48/51	50/54	BCG: 49/54	Qmax, PVR
**Iannucci, 2005 [25] Non-RCT**	265 (68/197)	NA	Transurethral resection	NA	ΝA	ΝA	ΝA	NA	NA	ΝA
**Jaidane, 2010 [26]** **Non-RCT**	170 (85/85)	71/71	Transurethral resection	pTa: 9/11 pT1: 76/74	G1: 32/33 G2: 45/44 G3: 8/8	70/65	15/20	35/33	BCG: 69/70	NA
**Kizilay, 2020 [27]** **Non-RCT**	186 (73/113)	62/48	Transurethral resection	pTa: 49/68 pT1: 24/36 pT2: 0/9	Low: 51/70 High: 22/43	62/89	11/24	NA	ΝA	NA
**Li, 2013 [28]** **RCT**	62 (25/37)	48/66	Photoselective vaporization	pTa: 3/6 pT1: 17/22 pT2: 5/9	Low: 20/28 High: 5/9	19/25	6/12	NA	BCG: NA, SIIC: all patients	NA
**Laor, 1981 [29]** **Non-RCT**	287 (137/150)	71/60	Transurethral resection	NA	G1: 51/58 G2: 35/35 G3: 51/57	112/124	25/26	69/96	NA	ΝA
**Li, 2013 [28]** **RCT**	62 (25/37)	48/66	Photoselective vaporization	pTa: 3/6 pT1: 17/22 pT2: 5/9	Low: 20/28 High: 5/9	19/25	6/12	NA	BCG: NA, SIIC: all patients	NA
**Park, 2009 [30]** **Non-RCT**	189 (24/165)	70/64	Transurethral resection	pTa/pTis: 8/56 pT1: 16/109	Low: 13/81 High: 11/84	NA	NA	52/44	BCG: 22/157	NA
**Singh, 2009 [31]** **RCT**	48 (24/24)	56/57	Transurethral resection	pTa: 17/18 pT1: 7/6	G1: 10/9 G2:11/11 G3: 3/4	24/24	0/0	36/38	NA	IPSS
**Ugurlu, 2007 [32]** **Non-RCT**	65 (31/34)	68/56	Transurethral resection	pTa: 25/25 pT1: 6/9	G1: 26/31 G2: 3/3 G3: 2/0	31/34	0/0	31/27	NA	ΝA
**Vicente, 1988 [33] Non-RCT**	200 (100/100)	69/60	Transurethral resection	pTa: 21/24 pT1: 79/76	G1: 4/18 G2: 78/73 G3: 18/9	58/52	42/48	47/46	None	ΝA
**Wang, 2020 [34]** **Non-RCT**	236 (118/118)	67/65	Thulium laser enucleation	pTa: 14/21 pT1: 114/97	Low: 71/79 High: 47/39	75/82	43/36	59/56	BCG: based on risk, SIIC: 89/93	Qmax, PVR

BC: bladder cancer; BCG: bacillus Calmette–Guérin; CG: control group; LUTS: lower urinary tract symptoms; NA: not available; Qmax: maximum flow rate; RCT: randomized controlled trial; SG: simultaneous group; SIIC: single intravesical instillation of chemotherapy; PVR: postvoid residual.

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
