# Peer review of "Concomitant Endoscopic Surgery for Bladder Tumors and Prostatic Obstruction: Are We Safely Hitting Two Birds with One Stone? A Systematic Review and Meta-Analysis"

_jcm, 2022, doi:10.3390/jcm11206208_

Round 1

Reviewer 1 Report

Thank you to the authors for this research work which, however,  is quite similar to a recent publication from BJUI 2021.

There are some important concerns about the review:

Major comments;

-        All tables and supplemental material should be shown for the review process. Is totally necessary to see the figures and supplemental material that has not been shared to be reviewed.

-        The authors should better justify the differences with previous recent metanalysis. They explain different hbp treatments but only two studies address this question and exploring different subtypes in terms of multifocality or grade again the number decrease significantly and the results do not vary in comparison with Motlagh and col which also include multifocal and high-grade tumours as shown in their table 2.…

-        It seems to not show any new strong evidence as the subgroup analysis or patient with postoperative chemotherapy are less than 400…

-        A summary of the review as an abstract should be shared.

-        In the introduction it would be better to give intervals percentage of the prevalence of HBP and symptoms by age and also CVNMI not only the recurrence or progression of this disease.

-        4-6% of patients with HBP or from those with primary CVNMI?

-        Supplementary Material 2 should be shown as Figure 1

-        Both PRISMA AND NOS are shown in the same supplemental material or is it a mistake?

-        Postoperative adjuvant therapy is poorly described and it is an important variable to assess the risk of recurrence.

Author Response

Reviewer #1

All tables and supplemental material should be shown for the review process. Is totally necessary to see the figures and supplemental material that has not been shared to be reviewed.

 Reply: Due to our oversight, we have not included our Supplementary material file. Please find attach our complete Supplementary with the following sections: Pubmed search string, Prisma flow diagram of study selection, Newcastle – Ottawa Scale for cohort studies, References of all excluded studies with reasons for exclusion, Quality assessment of included studies, Sensitivity analysis with high-quality studies, Subgroup analysis based on single intravesical instillation of chemotherapy, Analysis of time to first recurrence, Publication bias assessment, Subgroup analysis of recurrence based on grade of bladder tumor, Grading of evidence for all outcomes.

The authors should better justify the differences with previous recent metanalysis. They explain different hbp treatments but only two studies address this question and exploring different subtypes in terms of multifocality or grade again the number decrease significantly and the results do not vary in comparison with Motlagh and col which also include multifocal and high-grade tumours as shown in their table 2.

 Reply: We thank Reviewer #1 for his comment. We consider that the added value in the meta analysis by the different surgical approach overweight their small participants number, that is as correctly stated, an important limiting factor. We have included a statistical analysis of recurrence for patients underwent TURP vs different BPO modalities, mainly to detect, by proxy, if the well-known benefit of immediate intravesical instillations with chemotherapeutics can be also detected during simultaneous surgery. With the current literature and data in hand, no statistically significant difference can be detected, as illustrated by our analysis. As for the differences with the meta-analysis by Motlagh et al according the exploration of different subgroups (multifocality and Grade) , not only we share the patients descriptive, where ever provided, but also, as added compared to the aforementioned meta analysis, we have provided a comparative meta analysis between the different grades.(Supplement Data 9). Additionally, we provided sensitivity analysis of studies with good methodological quality, in order to provide a more robust result. All the corresponding forest plots are depicted in Data Supplement and the relevant findings are discussed in the Results section. Our systematic review and meta analysis covered a broader spectrum of the available literature, taking into consideration that we included 821 more cases in a total of 2421. Also we failed to locate another meta analysis incorporating also as outcome of interest the patients quality of life. In the end, according to the guidelines we have implemented GRADE into our findings defining their overall strength. The corresponding forest plots are depicted in Data Supplement 10 and the relevant findings are discussed in the Results section.

It seems to not show any new strong evidence as the subgroup analysis or patient with postoperative chemotherapy are less than 400…

 Reply: We thank Reviewer #1 for his comment. We are in line with this statement too, that no convincing evidence is been showed due to the low number of individuals in this subgroup analysis. Although, due to the importance of that cohort of patients -with postoperative chemotherapy- we felt that it would be interesting to detected if any, primitive, conclusions could be drawn. The results of this subgroup analysis are in line with the overall meta analysis results, that performing simultaneous transurethral prostatectomy with newer technologies using postoperative chemotherapy instillations is at least safe, as far oncological outcome is concerned, but more studies are needed in order to determine the level of postoperative intravesical instillations total impact.

A summary of the review as an abstract should be shared.

Reply: Due to our oversight, no abstract has been shared. Abstract is now in page 2 of our current submission.

In the introduction it would be better to give intervals percentage of the prevalence of HBP and symptoms by age and also CVNMI not only the recurrence or progression of this disease.

Reply:  We thank reviewer #1 for this constructive comment. Introduction has been modified in order to accumulate prevalence of BPO and NMIBC incidence rate. New citation was added. ( Egan KB. The Epidemiology of Benign Prostatic Hyperplasia Associated with Lower Urinary Tract Symptoms: Prevalence and Incident Rates. Urol Clin North Am. 2016 Aug;43(3):289-97. doi: 10.1016/j.ucl.2016.04.001. PMID: 27476122.)

4-6% of patients with HBP or from those with primary CVNMI?

Reply: We thank reviewer #1 for this constructive comment. We have added an explanation. 6% of patients underwent TUBT in the cited study have been found to have simultaneous BPH.

Supplementary Material 2 should be shown as Figure 1

Reply: We thank reviewer #1 for his suggestion. We have set Prisma flow diagram as Figure 1.

Both PRISMA AND NOS are shown in the same supplemental material or is it a mistake?

Reply: We thank reviewer #1 for his comment. It was an honest mistake. From now on Prisma flow diagram is Figure 1 and NOS is depicted as Data Supplement 4.

Postoperative adjuvant therapy is poorly described and it is an important variable to assess the risk of recurrence.

Reply: We thank reviewer #1 for his comment. In depth description of postoperative adjuvant therapy action has been added. New citation was added. ( Redelman-Sidi G, Glickman MS, Bochner BH. The mechanism of action of BCG therapy for bladder cancer—a current perspective. Nat Rev Urol. 2014 Mar;11(3):153-62. Doi: 10.1038/nrurol.2014.15. Epub 2014 Feb 4. PMID: 24492433.)

Reviewer 2 Report

Savvides at al in manuscript entitled "Concomitant endoscopic surgery for bladder tumors and prostatic obstruction, safely hitting two birds with one stone? a systematic review and meta-analysis"  assessed the effect of concurrent TURB and endoscopic BPO treatment on oncological safety.

The discussed topic is very interesting from a practical point of view and a meta-analysis of the existing literature was needed.

The work is well written and organized.

There are no methodological errors and all criteria for this type of article have been met.

Some minor aspects:

 - it would be interesting to refer to the latest work on this topic in the discussion - https://doi.org/10.1111/bju.15898

 - change Table 1 orientation to horizontal - much clearer view for readers.

Author Response

Reviewer #2

Savvides at al in manuscript entitled "Concomitant endoscopic surgery for bladder tumors and prostatic obstruction, safely hitting two birds with one stone? a systematic review and meta-analysis"  assessed the effect of concurrent TURB and endoscopic BPO treatment on oncological safety.The discussed topic is very interesting from a practical point of view and a meta-analysis of the existing literature was needed.The work is well written and organized.There are no methodological errors and all criteria for this type of article have been met.Some minor aspects:

 - it would be interesting to refer to the latest work on this topic in the discussion - https://doi.org/10.1111/bju.15898

 - change Table 1 orientation to horizontal - much clearer view for readers.

Reply: We thank Reviewer #2 for his compliments. We agree that a comment on the latest work on this subject will be useful. Comment in Discussion section has been added. New citation accordingly. Corrections to the Table 1 have been made in order to appear now horizontal.

Round 2

Reviewer 1 Report

Thank you to the authors for the revision.